# Health and Protective Measures for Seniors during the COVID-19 Pandemic in the Opinion of Polish Society

**DOI:** 10.3390/ijerph18179230

**Published:** 2021-09-01

**Authors:** Marta Podhorecka, Anna Pyszora, Agnieszka Woźniewicz, Jakub Husejko, Kornelia Kędziora-Kornatowska

**Affiliations:** 1Department of Geriatrics, Faculty of Health Sciences, Collegium Medicum in Bydgoszcz, Nicolaus Copernicus University, 85-094 Toruń, Poland; awozniewicz@cm.umk.pl (A.W.); kubahusejko@gmail.com (J.H.); kornelia.kornatowska@cm.umk.pl (K.K.-K.); 2Department of Palliative Care, Faculty of Health Sciences, Collegium Medicum in Bydgoszcz, Nicolaus Copernicus University, 85-094 Toruń, Poland; anna.pyszora@cm.umk.pl

**Keywords:** COVID-19, elderly, ageism, protective measures, vaccinations, FAQ

## Abstract

The aim of the study was to determine the opinion of society on the individual care and protection measures towards seniors during the COVID-19 pandemic. In addition, the relationship of opinions with demographic data, knowledge about aging and own experience in contacts with the elderly was examined. The study involved 923 attendees from Poland. The tools used to assess the research problem were: demographic characteristics, a Facts on Aging Quiz (FAQ), the author’s questionnaire about preventive and protective measures for seniors during the COVID-19 pandemic. We observed that over 50% of participants were against designating shopping hours for seniors. The analysis showed that negative attitudes were more often expressed by women than by men; younger people and those declaring that they do not spend too much time with the elderly. In the matter of vaccination priority for the elderly, over 70% participants replied “rather yes” or “definitely yes”. The use of the age criterion in situations of limited access to medications and ventilators was supported mainly by learners, with high results of the FAQ, and professionals dealing with seniors. Finally, almost 56% of participants declared that their contacts with seniors were the same as before the pandemic, while merely 1.6% indicated that they avoid contact with them entirely.

## 1. Introduction

A significant problem, both from the medical and socio-economic point of view, that emerged during the outbreak of the COVID-19 pandemic was the fact that elderly people are indicated as the most vulnerable to the severe, often fatal, course of COVID-19 [1]. This phenomenon was already mentioned in early reports from January 2020 from China [2,3] which allowed for a relatively quick development of strategies aimed at the most effective protection of the elderly against SARS-CoV-2. Due to the fact that the COVID-19 pandemic is a phenomenon on a global scale, the methods for protecting seniors had to be adapted to the economic and social possibilities of individual countries, which led to the formation of different strategies being developed and adopted for implementation [4].

As the last pandemic has shown, providing the elderly with special care could not always be introduced without the intensification of the negative phenomenon of ageism, which is defined by the WHO (2021) as the term used to describe the stereotypes, prejudice and discrimination against ourselves or other people, based on age [5,6]. The phenomenon of ageism has been described relatively recently, because it was not until 1969 that Robert Neil Butler called the term “ageism” as a discrimination of seniors, based on the previously described problems of sexism or racism [7], this issue is also not sufficiently publicized, and thus its existence is still not sufficiently present in the social consciousness [8]. In addition, its development was largely due to media campaigns and social campaigns that were carried out from the beginning of the pandemic. Words such as “vulnerable persons” or “elderly people” [9] were consistently used at that time, which means that people have been stereotyped in public discourse and on social media. Although they correctly indicated that older adults are more exposed to SARS-CoV-2 disease than younger people, but due to the lack of sufficient measures to prevent the social exclusion of seniors, it was often the case that elderly people were associated with the developing pandemic, and the responsibility for its development was passed on to them [10].

The high risk of the negative impact of the COVID-19 pandemic and the measures taken to protect seniors from the perception by other age groups justified conducting research to monitor the evolution of the phenomenon of ageism in individual countries. One of the countries, in which the conduct of this type of measurements is particularly important, is Poland, owing to the fact that it has a rapidly aging society [11] and the associated risk that the growing phenomenon of exclusion of the elderly may in the future develop into a major social problem in the country.

Conducting research on the opinions of Polish society on special activities carried out in relation to citizens of advanced age should be carried out on the basis of recalling the strategies already implemented. In this context, it is worth mentioning, first of all, the so-called “hours for seniors” introduced in Poland, on 14 October 2020, operating from Monday to Friday from 10.00 to 12.00 [12]. These were the hours when, by definition, younger social groups work or study, so this change was not supposed to interfere to a large extent in the everyday life of Poles, nevertheless, it was met with mixed reactions [13], and in some cases even with an unfounded recognition of people in the elderly category as a privileged group [14]. This situation could have a real impact on the perception of seniors now, but also in the future.

The rules for carrying out protective vaccinations against COVID-19, implemented as part of the National Immunization Program, are also important. This program assumes the immunization of elderly people, as they are particularly vulnerable to complications from SARS-CoV-2, earlier than younger people, if they are not medical personnel [15]. This strategy needs to be taken into account in studies on public opinion on the protection of older adults during the COVID-19 pandemic, as it may be a source of misinterpretation that seniors are a privileged group in society.

In taking measures to protect the elderly from the effects of the COVID-19 pandemic, it is also important to maintain the mental well-being of seniors, an important component of which is the fight against loneliness. This phenomenon is a significant problem in isolated people, especially if the isolation takes place without the presence of other people [16]. Methods of fighting loneliness among the elderly are sought in technological development, for example, in the form of media campaigns and programs aimed at older adults, especially due to the proven positive impact of such activities on the well-being of representatives of the social group in question [17]. The perception of such activities in society is also important, mainly in the context of assessing their effectiveness, which is why it is justified to conduct research on the evaluation of media campaigns targeting the elderly by various social groups.

The changes in everyday life that had to be introduced as a result of the COVID-19 pandemic became particularly noticeable at the turn of 2020 and 2021. First of all, a slight reduction in restrictions during the holiday months and the need to return to life with greater restrictions in autumn 2020, which were introduced due to the increase in the incidence of SARS-CoV-2 cases in this period [18], caused more and more people to feel frustrated with the prolonged pandemic. The bans on holding Christmas with the family, which is an important element of the Polish tradition, did not help to improve the public mood [19]. Increasing the restrictions in the fall, the ban on holding Christmas or the period of introducing hours for seniors were fresh memories for the inhabitants of Poland in February 2021 and could still influence their opinion. At the same time, it was a time when media programs and social campaigns targeted at people of advanced age had already established themselves, and where priority vaccination of elderly people under the National Immunization Program [15] began, and therefore, when seniors, associated with priority treatment, were most likely to become victims of the frustration of the younger part of society. The period of February 2021, therefore, seemed to be the best time to test how a series of rapidly successive events affecting everyday life could have a bearing on the perception of seniors in society.

In order to fully understand the public opinion of the actions taken to protect seniors from the effects of the COVID-19 pandemic, it is also necessary to ask questions specifying the relation between the respondents and the elderly in everyday life, which will allow the current interpersonal relations in Polish society to be defined and demonstrate what factors can influence opinions about older adults.

The aim of the study was to determine the opinion of the Polish society on individual care and protection measures towards seniors during the COVID-19 pandemic.

## 2. Materials and Methods

### 2.1. Purpose

Due to its sociodemographic characteristics, knowledge of aging and contact with older people in private and professional life, the detailed research questions were asked about the opinion of Polish society on particular care and protection measures towards seniors during the COVID-19 pandemic.

### 2.2. Participants

There was a cross-sectional internet-based survey that was conducted in the first week of February 2021. It was completed by 923 subjects, 448 men and 447 women. The entire Polish population is estimated at 38,265,000.

### 2.3. Measurements

The following research tools were used for the four surveys: demographic characteristics, Facts on Aging Quiz (FAQ), the original questionnaire about preventive and protective measures for seniors during the COVID-19 pandemic and the last group of questions, defining contact with the elderly.

The first questionnaire included demographic questions regarding the represented place of residence, marital status, education, professional situation, length of service, gross annual income.

The FAQ was used to define the knowledge about older age and was based on 25 statements [20,21,22]. Each of them had to be specified by the respondent as true or false. The tool contained statements that cover the guesswork about old age on the physical, psychological and social sides. For each correct answer, the participant received one point, the greater the number of correct answers, the wider the knowledge of the seniority team. The possible scores were 0–25 points.

The original questionnaire *health and protective measures for seniors during the COVID-19 pandemic*, consisted of five questions. The above questionnaire was developed by the research team. It was prepared on the basis of previous discussions with health care specialists (including geriatricians, epidemiologists, physiotherapists and psychologists working with the elderly), whom we asked to identify the most important issues related to health and protective measures for seniors during the COVID-19 pandemic. Then, the team conducted a qualitative pilot study, which allowed the questions to be verified, in terms of their understanding, readability and structure, on a group of 20 people of different sex, age and level of education. The first three questions could be answered on a five-point Likert scale (definitely yes, rather yes, I have no opinion, rather not, definitely not). The other two questions, due to their specificity, had to contain different answers (Table 1). One of the issues discussed were the hours in stores for seniors. Although they were already abolished during the study (they were in force in Poland from 31 March 2020–30 April 2020 and 15 October 2020–2 February 2021), they managed to arouse great controversy, and therefore prompted us to undertake this topic.

The last used tool was the original *contact with the elderly questionnaire*, consisting of 5 closed questions (Table 2).

### 2.4. Procedure and Data Collection

Participation was voluntary and anonymous. The study was based on the selection of a randomized sample, taking into account the criterion of age proportion for each group and gender, it was also a simple dependent sample. An online questionnaire was developed by the research team through the media and on the online platform (Survgo system), where the questionnaire itself was constructed. Only adults, with no upper age limit, took part in the study. All participants were informed that their responses were anonymous. Only fully completed questionnaires were entered into the analysis.

### 2.5. Data Analysis

We used R 4.0.2 (Bell Laboratories, Murray Hill, NJ, USA) to analyse the data [23]. To model the responses to the questions regarding policies related to elderly people during COVID-19 pandemic, a Bayesian ordinal regression was used, suitable for analyzing ordered categorical data [24]. In each of the analyses, the “no opinion” response was assumed to be equivalent to neutral attitude and constituted a midpoint of the scale. Ordered categorical predictors were coded with orthogonal linear contrast, unordered categorical predictors were coded with sum-to-zero contrast and continuous predictors were entered into a model on a standardized scale.

In Bayesian statistics, the inference is based on analyzing the posterior probability distributions of a model’s parameters (e.g., regression weights), obtained by integrating likelihood (data) with prior probability distributions. Regression weight is said to be statistically credible when 95% credible intervals (95% CI) of the posterior distribution exclude zero [25]. As a point estimate of the effect, the means of the posterior distributions are presented. Default improper flat priors were used for the regression weights.

Figures are presented to understand the relationship between a dependent variable and a predictor predicting marginal probabilities with corresponding 95% CI. These values represent a median of a posterior distribution of a proportion of people providing a given response predicted from the model parameters.

To approximate posterior distributions of the models, Markov Chain Monte Carlo (MCMC) sampling procedure was conducted using brms package [26]. Six parallel chains were used, each consisting of 8000 samples, with 4000 samples used as warm up period and every 10th sample recorded, resulting in 2400 recorded samples in total. Sampling procedure was efficient and resulted with well-mixed and not autocorrelated chains and unimodal posteriors. Model accuracy was assessed with posterior predictive checks.

## 3. Results

The characteristics of the participants are presented in Table 3. The raw proportions of responses to each of the analyzed questions is presented in Figure 1. We observed that over 50% of the participants were against designating shopping hours for elderly people. In the matter of vaccination priority for the elderly people, over 70% participants replied “rather yes” or “definitely yes”. For the questions “Age criteria by insufficient medication/respirator/etc. quantities during SARS-CoV-2 pandemic” over 40% people had no opinion, and responses no and yes accounted almost equally for the remaining 60% of the study sample. Over 45% of participants had no opinion regarding whether activating programs for elderly people should be organized during the pandemic, whereas almost 35% were in favor of such action. Finally, almost 56% of participants declared that their contacts with elderly people are the same as before the pandemic, while merely 1.6% indicated that they avoid contact with elderly people entirely. Next, we analyzed whether these responses were related to a set of demographic variables, FAQ regarding contacts with elderly people.

### 3.1. Designated Shopping Hours for Elderly People

The model coefficients of the ordinal regression with responses to questions about designated shopping hours for elderly people are summarized in Table 4, whereas the model predictions are presented in Figure 2. We observed three statistically credible relationships with responses to the questions. First, women tended to disagree with the policy more often, as the proportion of “definitely not” responses was much higher among them, as compared to the frequency of this response among men. Second, agreeableness to the policy increases with age, as indicated by credibly decreasing proportions of “definitely not” answers to the questions as the age of the respondents increased, while the proportions of “rather yes” and “definitely yes” responses increased with age. Finally, responses to the questions were also credibly and strongly related to the relevance of contacts with elderly people. As the relevance increased, the proportion of responses “definitely not” strongly decreased, and proportions of responses “rather yes” and “definitely yes” slightly increased.

### 3.2. Vaccination Priority for Elderly People

Model coefficients of the ordinal regression with responses to questions about SARS-CoV-2 vaccination priority for elderly people are summarized in Table 5, whereas model predictions are presented in Figure 3. We observed three statistically credible relationships with responses to the questions. First, as the age of the participants increased, the proportion of “definitely yes” responses credibly and strongly increased, and proportions of “definitely not” and “rather not” responses slightly increased. Second, as the knowledge of FAQ increased, the proportion of “definitely yes” answers credibly decreased. Finally, as the importance of contacts with elderly people increased, the proportion of “definitely yes” responses strongly increased, and proportions of “definitely not” and “rather not” answers slightly decreased.

### 3.3. Age Criteria for Medication/Respirator/etc. Rationing during SARS-CoV-2 Pandemic

The model coefficients of the ordinal regression with responses to age criteria for medication/respirator/etc. rationing during SARS-CoV-2 pandemic* are summarized in Table 6, whereas model predictions are presented in Figure 4. We observed three statistically credible relationships with responses to the questions. First, professional status was credibly related to answers to the questions, with the highest proportion of “yes” answers among “at school” participants and the lowest proportion of “yes” answers among unemployed participants. Second, the proportion of no answers credibly decreased, and proportion of the answers credibly increased, as the knowledge of FAQ increased. Finally, having professional contacts with elderly people also was credibly related to the responses to the questions, with the highest proportion of “yes” answers. * Fifty-six “other” answers to this question were not included in the analysis.

### 3.4. Therapeutic Activation Programs for Elderly People

The model coefficients of the ordinal regression with responses to questions regarding therapeutic activation programs for elderly people during the COVID-19 pandemic are summarized in Table 7, whereas the model predictions are presented in Figure 5. We observed five statistically credible relationships with responses to the questions. First, marital status was credibly related to answers to the questions, with the highest proportions of “rather yes” and “definitely yes” answers among married participants, and the lowest proportions of these two answers among single participants. Second, professional status was credibly related to answers to the questions, with the lowest proportions of “rather yes” and “definitely yes” answers among employed participants. Third, the proportions of “rather yes” and “definitely yes” answers credibly increased, and the proportions of “rather not” and “definitely not” answers credibly decreased as the knowledge of FAQ increased. Fourth, having professional contacts with elderly people also was credibly related to the responses to the questions, with the highest proportion of “rather yes” and “definitely yes” answers among people who have such contacts with elderly people. Finally, the proportions of “rather yes” and “definitely yes” answers credibly increased, and the proportions of “rather not” and “definitely not” answers credibly decreased as the relevance of contacts with elderly people increased.

### 3.5. Contacts with Elderly People

The model coefficients of the ordinal regression with responses to questions regarding contacts with elderly people during the COVID-19 pandemic are summarized in Table 8, whereas the model predictions are presented in Figure 6. We observed five statistically credible relationships with responses to the questions. First, gender was credibly related to the pattern of contact with elderly people, with higher proportions of “same as before” answers and lower proportions of “reduce to some degree/significantly” answers among males than among females. Second, the proportion of “same as before” answers credibly decreased, and proportions of “reduce to some degree/significantly” answers credibly increased, as the age of the respondents increased. Third, marital status was credibly related to answers to the questions, with the lowest proportion of “same as before” and highest proportion of “reduce significantly” answers among people in informal relationships. Fourth, as frequency of personal contacts with elderly people increased, the proportion of “same as before” answers credibly increased, and proportions of “reduce to some degree/significantly” answers decreased. Finally, keeping contacts with elderly people from outside the family was also credibly related to the answers to the questions.

## 4. Discussion

Acknowledging that the elderly and persons with underlying health conditions are more susceptible to COVID-19, a growing number of stores around the world were dedicating time or opening earlier for senior shoppers [5,27]. Special shopping hours, dedicated to older adults who are among the most vulnerable to severe complications from COVID-19, were set in Poland by government decision from 10 to 12. Thus, only people over 60 years of age could do shopping at this time in Poland. In our study, 50% of the respondents were against setting shopping hours for the elderly. Women more often than men did not agree with this policy. Policy compliance increased with age and was strongly associated with the relevance of contacts with the elderly. As the practice of social distancing became the norm, the special hours at supermarkets could further limit coronavirus exposure for older shoppers who chose to go out, which put them more at risk of infection. Nevertheless, appointing hours only for seniors aroused various emotions in many Poles. Not everyone was satisfied with this state of affairs, especially that most of the seniors could ask family, friends or neighbors to do shopping for them.

With most of the world under quarantine, the elderly are somewhat more vulnerable than younger ones. They have more chronic conditions than young subjects. Their aging immune systems makes it harder to fight off diseases, infections and viruses and recovery is usually slower and more complicated. For these reasons, contact with seniors may be associated with concerns about their health, the possibility of infecting them, which causes stress. Social distancing to protect older adults from COVID-19 infection can inadvertently increase loneliness, depression, health problems, and negative stereotyping of seniors⁵. In our study almost 56% of the respondents declared that their contacts with the elderly were the same as before the pandemic, and only 1.6% indicated that they completely avoided contact with the elderly. Repeated caution of high risk of potentially fatal complications if infected with COVID-19 virus increased a sense of helplessness and anxiety in both older and younger people. A 65-year-old male from India is an example. He had been hospital quarantined after testing positive for COVID-19, experienced symptoms of panic and anxiety. The patient blamed everyone around him for infecting him [28]. Another study confirmed that the pandemic was associated with changes in mental health. The 18.7% of the Spanish sample revealed depressive, 21.6% anxiety and 15.8% PTSD symptoms [29].

Aspects of social relationships were related to loneliness both before and during the pandemic in older Swiss adults [30]. While physical distance is useful for infection prevention and control, social isolation through limited interactions can negatively impact the cognitive, mental and physical functions of older adults [31].

According to the National Immunization Program, elderly people in Poland were vaccinated against SARS-CoV-2 immediately after medical personnel, as people particularly vulnerable to the consequences of the COVID-19 pandemic [15]. According to our research, as the age of the respondents increased, the percentage of “definitely yes” answers increased significantly, which can be explained by the fact that this priority may mean that respondents who are more advanced in age have a better chance of receiving a vaccination sooner. This reveals the phenomenon of high demand for vaccination, far exceeding the supply, which is a problem on a global scale. An example confirming the high demand for vaccination, and at the same time emphasizing the determination of those interested, was the analysis carried out in Ecuador, according to which at least 85% of the respondents were able to pay extra for receiving the vaccination [32]. High demand, although it was found to decrease over the course of the pandemic, was also found in the National Serial Study conducted in Kuwait [33]. There are no similar studies that have been conducted in Polish society, but the results we obtained suggest that the level of determination and actions aimed at obtaining a vaccine for oneself may be high.

Another fact observed in our study, in which the percentage of answers “definitely yes” decreased along with the increase in knowledge about aging, suggests that people with the greatest knowledge about aging processes do not treat them as sufficient for people at the time of aging to need vaccinating more than other age groups. It is an observation that undoubtedly requires further analysis and research, especially since the knowledge at the time after one year of the pandemic clearly indicates that elderly people are particularly at risk due to COVID-19, and their early vaccination is fully justified [34]. Different from the available knowledge, the position of people who know the aging processes may mean the presence of a very unfavorable phenomenon, in which even people with the appropriate amount of knowledge are not aware of the risks to which seniors are exposed.

It was possible to predict the dependencies, according to which with the growing importance of contacts with the elderly, the percentage of answers “definitely yes” increased significantly, and “definitely no” and “probably not” decreased. These results indicate that people attached to the elderly are aware of the need to protect seniors from the consequences of the COVID-19 pandemic. Taking into account the fact that knowledge about vaccination against SARS-CoV-2 is constantly developing [35], this may mean the growth of a positive phenomenon in which the awareness of the very large impact of vaccination on the health of the elderly will increase, and people who will be particularly alarmed will be those for whom the well-being of seniors is extremely important.

Already in the spring of 2020, the high number of COVID-19 cases made many physicians face a very difficult choice related to the need to prioritize access to selected forms of treatment (e.g., mechanical ventilation) [36]. In our study, we asked respondents about the legitimacy of using the age criterion when determining the priority rules for selected forms of treatment. Over 40% of respondents indicated that they did not have an opinion on this subject. The high number of responses, showing the lack of an unambiguous position on this matter, may indicate how complex the problem is that we are dealing with. Interestingly, in the group of people who were able to express an opinion, high rates of knowledge about the facts related to the aging process were observed. Among the respondents who expressed their opinion (60%), the votes “for” using the age criterion and the votes “against” were evenly distributed. A high percentage of responses for the priority of younger people’s access to selected, deficit treatment methods concerned students. What is very disturbing, the results of our survey indicate that people who have contacts with the elderly, in connection with their professional work, indicated responses in favor of the access of younger people to selected, deficient methods of treatment. This confirms the attitudes of social discrimination based on age observed in many societies [8,37]. Unfortunately, the age criterion in the priority of access to limited resources also appears in the guidelines for the allocation of ventilators adopted in some countries [38]. Many such provisions are controversial from the point of view of medical ethics, which indicates that the most important decision-making criteria are the need for therapy and the forecasted chances of survival. The recorded age is not an objective measure of the latter.

In many countries, since the outbreak of the pandemic, there have been a number of initiatives aimed at activating the elderly, who very often remain at home in order to stay safe. Lack of activity adversely affects their physical and mental health [38,39]. More than 1/3 of people included in our study expressed a positive opinion on the programs aimed at activating the elderly. The highest percentage of answers “for” concerned married people, while the lowest was among single people. Importantly, the higher the knowledge about the FAQ, the more often the participants of the study positively assessed programs dedicated to the elderly. This clearly shows the importance of social education on the aging process, as it has great potential in shaping “sensitive” attitudes to the needs of seniors. The results of our analysis also indicate that people who have contacts with the elderly as part of their professional work expressed unambiguously positive opinions about the need to implement programs for activation of the elderly. Importantly, positive responses significantly increased along with the importance of contacts with the elderly by the respondents. The more important these contacts were, the greater the number of respondents declared their strong support for the activation programs for seniors. Therefore, it seems that the basis of the strategy minimizing the phenomenon of ageism is, on the one hand, social education aimed at learning the facts related to the aging process, and on the other hand, enabling social relations between people of different ages, which may translate into an improvement in mutual understanding, and hence limiting the attitudes discriminating against the elderly [40,41]. In further studies, it would be worth considering if the relationship with FAQ and vaccine prioritization opinions is dependent on other factors. The findings of the presented study may be important for theory and practice to be undertaken for seniors in situations where they are a group exposed to difficulties in contact with others, which may significantly affect their quality of life.

## 5. Limitations of the Study

There were several limitations to this study. First of all, in the own research presented here was its implementation through an online questionnaire. It is related to the existing epidemic situation in Poland and the desire not to expose society to the SARS-CoV-2 infection. In the future, the study group should be expanded to include people who do not have access to the Internet and the results obtained should be confronted. The presented study as well as the majority of the studies cited were run at the beginning of pandemic. It would be worthwhile to extend the research to the next stages, after the first, second and third waves of the pandemic, as well as to check whether opinion changed with the successive stages. Future studies should also endeavor to include a larger sample of participants. The authors hope that future research will focus on active ways to reduce or eliminate ageism, not only in the context of a pandemic, which was omitted in presented research.

## 6. Conclusions and Considerations for Implementation

As part of the research we have carried out, the following conclusions can be drawn:Regarding the priority of vaccination for the elderly, the majority (over 70%) of participants were in favor of such measures. The support of the majority of Polish society in this matter is of great importance for the policy related to immunization of the population against COVID-19 in Poland, because it indicates that the decision to vaccinate the elderly was met with a good public response and may be repeated if the need occurs for revaccination of the population range. Similar research should be carried out in other countries to determine how the priority of seniors, in terms of vaccination, is perceived in a global perspective.Half of the participants were against setting shopping hours for the elderly, while the survey showed that the support for such activities grew with the age of the respondents. The results obtained in this way clearly indicate that the support or lack thereof for the mentioned idea is related not to the phenomenon of discrimination against the elderly, but to the presence or absence of individual benefits related to the implementation of the solution. This is evidenced by the results showing the growing support for the idea with age, which suggests that the respondents were rather guided by their own benefits related to scheduling shopping hours for the elderly. However, taking into account that as the COVID-19 pandemic has progressed, a noticeable departure from the described solution has been made, and the obtained results are probably not related to ageism, this topic should not be considered important in the context of research on the phenomenon of social exclusion of seniors.In the case of the implementation of activation programs for seniors during the pandemic, almost half of the respondents did not have an opinion on this subject, but the group declaring support (35%) for such activities was characterized by greater knowledge about aging. It should be concluded here that when conducting policies aimed at reducing the phenomenon of ageism in Poland and other countries, campaigns to educate society should be particularly important, because our research clearly indicates that a high level of knowledge about aging also entails greater awareness of activities leading to the improvement of the quality of seniors’ lives, especially in the conditions of pandemic. Such educational activities would allow in the longer term to introduce further, more advanced solutions limiting the phenomenon of discrimination against people in advanced age.The most even distribution of opinions was observed when the “age criterion by insufficient drugs/respirator/etc. amounts during the SARS-CoV-2 pandemic”, where more than 40% of respondents had no opinion, and the answers “yes” and “no” came out with almost equal frequency. However, it was noticed that among people with more knowledge about aging, hesitancy was more often indicated in the response, and among learners, favoring such activities was declared more often. It follows that in subsequent studies in the field of ageism, it is worth measuring the ratio of the elderly in those who care for seniors professionally, but also in learners. Greater support for the above-mentioned solutions among the second group is an interesting phenomenon in the context of healthcare organization and suggests a lower discriminatory attitude among younger people who will work with seniors in the future. However, our research is not sufficient to draw such conclusions and further research on the described phenomenon will be needed.More than half of the participants maintain the same contact with the elderly as before the pandemic, while just over 1% indicate that they completely avoid contact with them. However, in the presented research, there is a noticeable reduction in contacts in the case of an increase in the age of the respondents. This information is also important in the context of policies aimed at limiting the phenomenon of ageism. Considering the fact that limiting contact with exposed people during the COVID-19 pandemic was not satisfactory, educational campaigns focusing mainly on the risks associated with the spread of the pandemic should be considered. Particular care should be taken to avoid the negative phenomenon of stereotyping people in old age as a potential source of the spread of the coronavirus.An important element of the study was the contact with the elderly, where it was noticed that with the increasing importance of contacts, the need to implement hours for seniors, priority in vaccinations and insufficient availability of activation programs for seniors during the pandemic was observed more often. This is a valuable indication in the context of conducting a policy aimed at reducing the phenomenon of discrimination against seniors in society. According to the obtained results, actions leading to the integration of seniors with younger people, at first educating and then introducing actual integration, may be positive solutions. Actions, such as the results of our study suggest, will sensitize younger people to the needs of seniors and encourage them to take action to improve their quality of life and effectively reduce the phenomenon of ageism.

In sum, our findings add to the knowledge information of societal attitudes on aging. An important implication of this study is that to improve people’s agreement with policies to reduce ageism and to reduce blame on older adults for the pandemic in Poland awareness-raising campaigns should be run, as well as disseminating information to the entire population by publishing it in publicly available sources.

## Figures and Tables

**Figure 1 ijerph-18-09230-f001:**
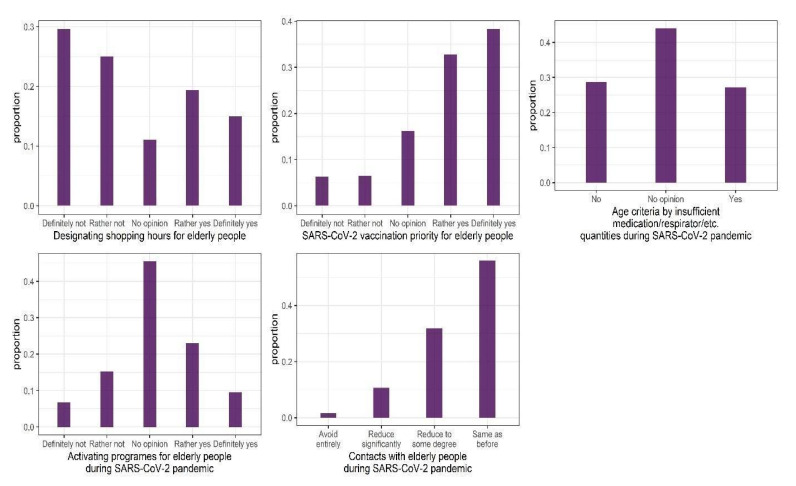
Distributions of responses to the questions regarding policies related to elderly people during COVID-19 pandemic.

**Figure 2 ijerph-18-09230-f002:**
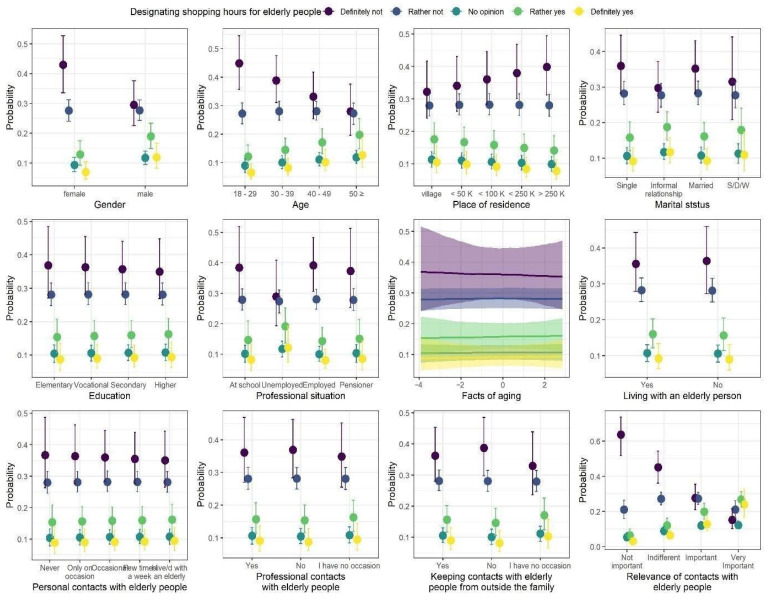
Posterior medians (points) of the predicted proportions of each response to the question regarding designated shopping hours for elderly people. The proportions sum to one within levels of each predictor. Vertical lines are 95% credibility intervals.

**Figure 3 ijerph-18-09230-f003:**
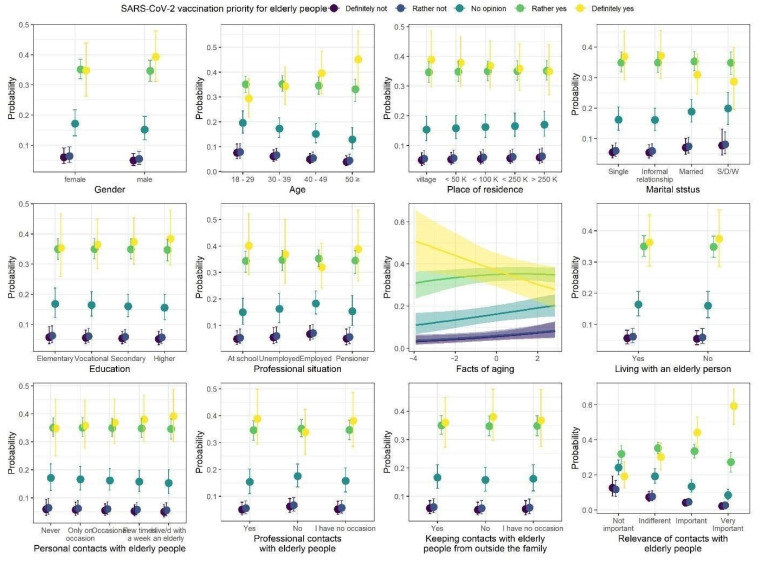
Posterior medians (points) of the predicted proportions of each response to the question about SARS-CoV-2 vaccination priority for elderly people. The proportions sum to one within levels of each predictor. Vertical lines are 95% credibility intervals.

**Figure 4 ijerph-18-09230-f004:**
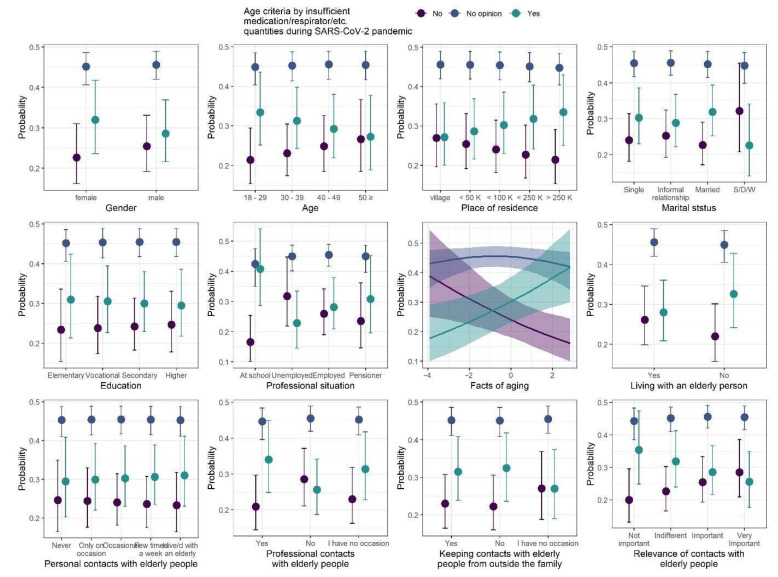
Posterior medians (points) of the predicted proportions of each response to age criteria for medication/respirator/etc. rationing during SARS-CoV-2 pandemic. The proportions sum to one within levels of each predictor. Vertical lines are 95% credibility intervals.

**Figure 5 ijerph-18-09230-f005:**
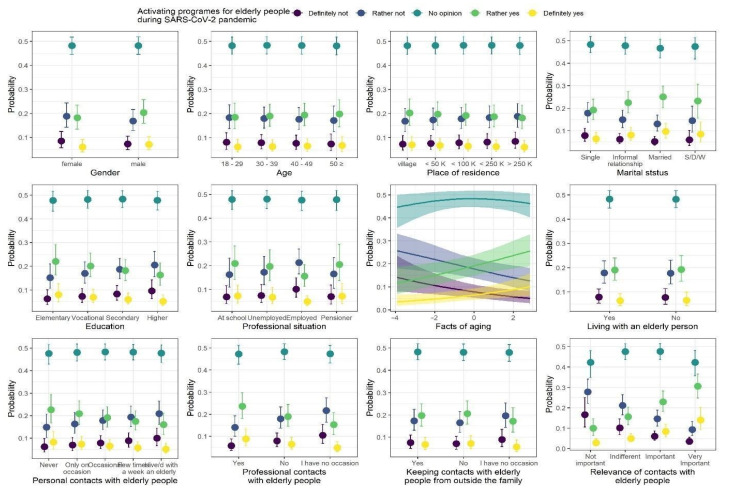
Posterior medians (points) of the predicted proportions of each response to the question regarding therapeutic activation programs for elderly people during COVID-19 pandemic. The proportions sum to one within levels of each predictor. Vertical lines are 95% credibility intervals.

**Figure 6 ijerph-18-09230-f006:**
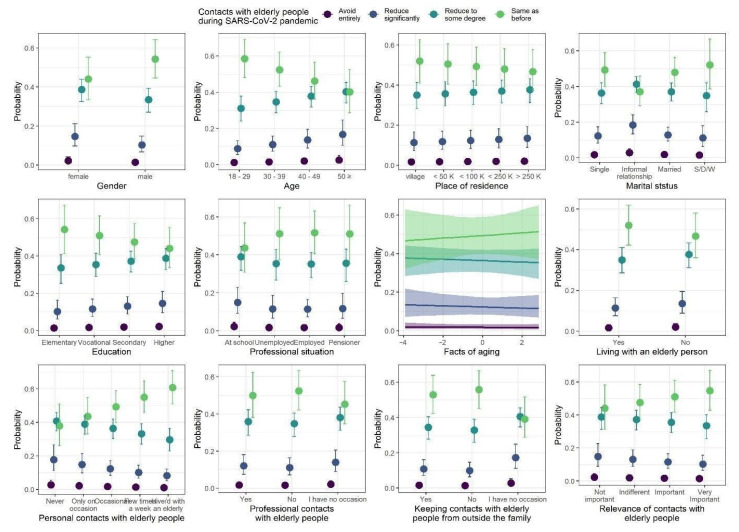
Posterior medians (points) of the predicted proportions of each response to the question regarding contacts with elderly people during COVID-19 pandemic. The proportions sum to one within levels of each predictor. Vertical lines are 95% credibility intervals.

**Table 1 ijerph-18-09230-t001:** The original questionnaire: health and protective measures for seniors during the COVID-19 pandemic.

Health and Protective Measures for Seniors during the COVID-19 Pandemic Questionnaire		
1	Do you think it is right that during the SARS-CoV-2 pandemic, there were designated hours in which only seniors could make purchases (the so-called “senior hours”)	(a) Definitely yes (b) Rather yes (c) I have no opinion (d) Rather not (e) Definitely not
2	Do you think it is right that during the implementation of the National Immunization Program (SARS-CoV-2), the elderly are/were/will be vaccinated first after healthcare professionals?	(a) Definitely yes (b) Rather yes (c) I have no opinion (d) Rather not (e) Definitely not
3	In your opinion, were the programs activating socially active elderly people (TV, Internet) conducted during the SARS-CoV-2 pandemic a sufficient replacement for traditional activating actions?	(a) Definitely yes (b) Rather yes (c) I have no opinion (d) Rather not (e) Definitely not
4	Do you think that the use of the age criterion in the case of insufficient number of drugs/ventilators/intensive care units during treatment with SARS-CoV-2 and their transfer to the younger part of society is justified?	(a) Yes (b) No (c) I have no opinion (d) I think that additional criteria should still be taken into account. If so, what are they?
5	Are you afraid of contact with the elderly, so as not to expose them to SARS-CoV-2 infection?	(a) No, my contacts with the elderly are the same as before the pandemic (b) I limit contact to some extent (e.g., shorten meetings), apply additional protective measures and increase the distance (c) I significantly limit my contact with the elderly (d) I completely avoid meeting with the elderly

**Table 2 ijerph-18-09230-t002:** The original questionnaire “Contact with the elderly”.

Contact with the Elderly—Questionnaire		
1	Do you live or have you ever lived with an elderly person?	(a) Yes (b) No
2	Do you have contact with the elderly in your private life:	(a) Yes, I live with an elderly person (b) Yes, several times a week (c) Only at ceremonies (d) Occasional (e) Not at all
3	Do you have contact with people over 65 at work?	(a) Yes (b) No (c) I don’t have the opportunity
4	Are you in contact with elderly people who are not family members?	(a) Yes (b) No (c) I don’t have the opportunity
5	How important are contacts with the elderly to you?	(a) Very important (b) Important (c) It doesn’t matter (d) They are not important

**Table 3 ijerph-18-09230-t003:** Characteristics of participants (n = 923).

Gender	N	%
female	475	51.46
male	448	48.54
**Age**	**N**	**%**
18–29	251	27.19
30–39	233	25.24
40–49	234	25.35
50≥	205	22.21
**Place of residence**	**N**	**%**
village	163	17.66
<50 K	207	22.43
<100 K	159	17.23
<250 K	138	14.95
>250 K	256	27.74
**Marital status**	**N**	**%**
Single	158	17.12
Informal relationship	225	24.38
Married	482	52.22
S/D/W	58	6.28
**Education**	**N**	**%**
Elementary	28	3.03
Vocational	98	10.62
Secondary	407	44.1
Higher	390	42.25
**Professional situation**	**N**	**%**
At school	73	7.91
Unemployed	81	8.78
Employed	686	74.32
Pensioner	83	8.99
**Annual income**	**N**	**%**
0–20,999	153	16.58
21,000–40,999	220	23.84
41,000–60,999	215	23.29
61,000–80,999	151	16.36
81,000 and more	117	12.68
Refuse to answer	67	7.26

**Table 4 ijerph-18-09230-t004:** Results of Bayesian ordinal regression with responses to the questions about designated shopping hours for elderly people as dependent variable.

	β	SE	LI	UI
**Gender (female—male)**	**−0.3**	**0.07**	**−0.43**	**−0.16**
**Age**	**0.55**	**0.18**	**0.21**	**0.9**
Place of residence	−0.26	0.14	−0.53	0.01
Marital status (Informal relationship)	0.28	0.19	−0.1	0.66
Marital status (Married)	0.03	0.19	−0.34	0.43
Marital status (S/D/W)	0.2	0.3	−0.4	0.77
Education	0.06	0.19	−0.32	0.43
Professional status (at achool)	−0.11	0.21	−0.51	0.29
Professional status (unemployed)	0.31	0.18	−0.04	0.66
Professional status (employed)	−0.14	0.13	−0.39	0.11
FAQ *	0.01	0.06	−0.12	0.13
Living with an elderly person (Yes/no)	0.02	0.08	−0.15	0.17
Personal contacts with elderly people	0.06	0.21	−0.33	0.5
Professional contacts with elderly people (yes)	−0.01	0.1	−0.2	0.2
Professional contacts with elderly people (no)	−0.04	0.09	−0.22	0.14
Keeping contacts with elderly people from outside the family (yes)	−0.01	0.09	−0.19	0.18
Keeping contacts with elderly people from outside the family (no)	−0.12	0.1	−0.32	0.09
**Relevance of contacts with elderly people**	**1.72**	**0.23**	**1.26**	**2.18**
Model accuracy	0.35			

Note: β and SE are posterior mean and standard error of the mean, respectively. LI and UI are lower and upper boundaries of the 95% credibility interval. Bolded rows indicate statistically credible regression weights. Model accuracy is the average proportion (across possible responses to questions) of correct response predictions from the model. * Facts on Aging Quiz.

**Table 5 ijerph-18-09230-t005:** Results of Bayesian ordinal regression with responses to the questions about SARS-CoV-2 vaccination priority for elderly people.

	β	SE	LI	UI
Gender	−0.1	0.08	−0.25	0.05
**Age**	**0.51**	**0.18**	**0.17**	**0.85**
Place of residence	−0.13	0.14	−0.4	0.14
Marital status [1]	0	0.19	−0.38	0.38
Marital status [2]	−0.27	0.19	−0.62	0.09
Marital status [3]	−0.38	0.3	−0.97	0.19
Education	0.08	0.2	−0.31	0.47
Professional status [1]	0.13	0.2	−0.27	0.53
Professional status [2]	−0.01	0.19	−0.37	0.35
Professional status [3]	−0.22	0.13	−0.47	0.05
**FAQ ***	**−0.15**	**0.06**	**−0.27**	**−0.02**
Living with an elderly person	−0.02	0.08	−0.17	0.13
Personal contacts with elderly people	0.15	0.22	−0.26	0.58
Professional contacts with elderly people [1]	0.09	0.11	−0.12	0.31
Professional contacts with elderly people [2]	−0.14	0.09	−0.32	0.03
Keeping contacts with elderly people from outside the family [1]	−0.05	0.1	−0.24	0.14
Keeping contacts with elderly people from outside the family [2]	0.05	0.1	−0.16	0.25
**Relevance of contacts with elderly people**	**1.35**	**0.24**	**0.91**	**1.8**
Model accuracy	0.41			

Note: β and SE are posterior mean and standard error of the mean, respectively. LI and UI are lower and upper boundaries of the 95% credibility interval. The [n] symbol indicates nth coefficient of a sum-to-zero contrast for a categorical predictor. Bolded rows indicate statistically credible regression weights. Model accuracy is the average proportion (across possible responses to questions) of correct response predictions from the model. * Facts on Aging Quiz.

**Table 6 ijerph-18-09230-t006:** Results of Bayesian ordinal regression with responses to age criteria for medication/respirator/etc. rationing during SARS-CoV-2 pandemic.

	β	SE	LI	UI
Gender	0.08	0.08	−0.08	0.24
Age	−0.22	0.19	−0.58	0.16
Place of residence	0.24	0.14	−0.04	0.52
Marital status [1]	−0.07	0.2	−0.46	0.33
Marital status [2]	0.08	0.2	−0.3	0.48
Marital status [3]	−0.4	0.33	−1.02	0.23
Education	−0.05	0.2	−0.44	0.36
**Professional status [1]**	**0.46**	**0.22**	**0.03**	**0.89**
**Professional status [2]**	**−0.39**	**0.18**	**−0.74**	**−0.04**
**Professional status [3]**	**−0.1**	**0.13**	**−0.36**	**0.16**
**FAQ ***	**0.18**	**0.07**	**0.05**	**0.31**
Living with an elderly person	−0.11	0.08	−0.28	0.05
Personal contacts with elderly people	0.06	0.22	−0.39	0.49
**Professional contacts with elderly people [1]**	**0.18**	**0.11**	**−0.04**	**0.38**
**Professional contacts with elderly people [2]**	**−0.23**	**0.1**	**−0.42**	**−0.04**
Keeping contacts with elderly people from outside the family [1]	0.06	0.1	−0.13	0.25
Keeping contacts with elderly people from outside the family [2]	0.1	0.11	−0.11	0.32
Relevance of contacts with elderly people	−0.35	0.24	−0.81	0.12
Model accuracy	0.46			

Note: β and SE are posterior mean and standard error of the mean, respectively. LI and UI are lower and upper boundaries of the 95% credibility interval. The [n] symbol indicates nth coefficient of a sum-to-zero contrast for a categorical predictor. Bolded rows indicate statistically credible regression weights. Model accuracy is the average proportion (across possible responses to a question) of correct response predictions from the model. * Facts on Aging Quiz.

**Table 7 ijerph-18-09230-t007:** Results of Bayesian ordinal regression with responses to the questions regarding therapeutic activation programs for elderly people during COVID-19 pandemic.

	β	SE	LI	UI
Gender	−0.08	0.07	−0.23	0.06
Age	0.08	0.18	−0.26	0.42
Place of residence	−0.13	0.14	−0.4	0.16
**Marital status [1]**	**0.25**	**0.2**	**−0.15**	**0.62**
**Marital status [2]**	**0.43**	**0.19**	**0.05**	**0.79**
**Marital status [3]**	**0.27**	**0.3**	**−0.32**	**0.83**
Education	−0.34	0.19	−0.71	0.05
**Professional status [1]**	**0.13**	**0.21**	**−0.27**	**0.54**
**Professional status [2]**	**0.04**	**0.18**	**−0.31**	**0.4**
**Professional status [3]**	**−0.29**	**0.13**	**−0.54**	**−0.04**
**FAQ ***	**0.17**	**0.07**	**0.04**	**0.3**
Living with an elderly person	0	0.08	−0.15	0.15
Personal contacts with elderly people	−0.42	0.22	−0.83	0.02
**Professional contacts with elderly people [1]**	**0.33**	**0.11**	**0.12**	**0.54**
**Professional contacts with elderly people [2]**	**−0.01**	**0.09**	**−0.19**	**0.17**
Keeping contacts with elderly people from outside the family [1]	0.04	0.1	−0.15	0.22
Keeping contacts with elderly people from outside the family [2]	0.12	0.11	−0.09	0.33
**Relevance of contacts with elderly people**	**1.28**	**0.23**	**0.83**	**1.73**
Model accuracy	0.45			

Note: β and SE are posterior mean and standard error of the mean, respectively. LI and UI are lower and upper boundaries of the 95% credibility interval. The [n] symbol indicates nth coefficient of a sum-to-zero contrast for a categorical predictor. Bolded rows indicate statistically credible regression weights. Model accuracy is the average proportion (across possible responses to questions) of correct response predictions from the model. * Facts on Aging Quiz.

**Table 8 ijerph-18-09230-t008:** Results of Bayesian ordinal regression with responses to the questions regarding contacts with elderly people during COVID-19 pandemic.

	β	SE	LI	UI
**Gender**	**−0.21**	**0.08**	**−0.37**	**−0.07**
**Age**	**−0.56**	**0.2**	**−0.93**	**−0.18**
Place of residence	−0.16	0.15	−0.46	0.12
**Marital status [1]**	**−0.5**	**0.22**	**−0.93**	**−0.07**
**Marital status [2]**	**−0.06**	**0.22**	**−0.49**	**0.36**
**Marital status [3]**	**0.12**	**0.34**	**−0.55**	**0.81**
Education	−0.31	0.21	−0.72	0.1
Professional status [1]	−0.23	0.22	−0.66	0.2
Professional status [2]	0.08	0.19	−0.27	0.45
Professional status [3]	0.09	0.14	−0.18	0.37
FAQ *	0.03	0.07	−0.11	0.16
Living with an elderly person	0.1	0.08	−0.06	0.27
**Personal contacts with elderly people**	**0.74**	**0.23**	**0.3**	**1.2**
Professional contacts with elderly people [1]	0.03	0.12	−0.21	0.25
Professional contacts with elderly people [2]	0.13	0.1	−0.06	0.32
**Keeping contacts with elderly people from outside the family [1]**	**0.15**	**0.1**	**−0.06**	**0.35**
**Keeping contacts with elderly people from outside the family [2]**	**0.26**	**0.11**	**0.04**	**0.49**
Relevance of contacts with elderly people	0.32	0.26	−0.18	0.82
Model accuracy	0.57	0.57	0.57	0.57

Note: β and SE are posterior mean and standard error of the mean, respectively. LI and UI are lower and upper boundaries of the 95% credibility interval. The [n] symbol indicates nth coefficient of a sum-to-zero contrast for a categorical predictor. Bolded rows indicate statistically credible regression weights. Model accuracy is the average proportion (across possible responses to a question) of correct response predictions from the model. * Facts on Aging Quiz.

## Data Availability

The data presented in this study is available from the respective author upon request. The data is not publicly available due to the continuation of the project in this regard.

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
