# Peer review of "Health and Protective Measures for Seniors during the COVID-19 Pandemic in the Opinion of Polish Society"

_ijerph, 2021, doi:10.3390/ijerph18179230_

Round 1
Reviewer 1 Report
Thank you for the opportunity to review the manuscript "Health and protective measures for seniors during the covid-19 pandemic in the opinion of polish society".
In my opinion, this is a novel and interesting study. I have no major comment, and only minor comments are shown below.
1) Figure 2,Figure 3 and Figure 4 in the manuscript were placed in inappropriate positions, which affects the review experience and should be adjusted.
2) Figure 5 mentioned in line 287 does not exist in the manuscript and must be added when revising.
3) The section of concluding and considerations for implementation in lines 459 to 491 mainly repeated the results of data analysis and lacked considerations for implementation, especially the considerations of the value of these results for policy implementation, which needs to be rewritten in the revised version.
Author Response
Response to Reviewer 1 Comments
Dear Reviewer,
We very much appreciate the careful review of our manuscript titled “Health and protective measures for seniors during the covid-19 pandemic in the opinion of polish society”. We have studied your comments carefully and made major correction which we hope meet with your approval. We answer your questions or comments in details in the following texts.
Points:
- Figure 2,Figure 3 and Figure 4 in the manuscript were placed in inappropriate positions, which affects the review experience and should be adjusted.
- We made a correction
- Figure 5 mentioned in line 287 does not exist in the manuscript and must be added when revising.
- We made a correction
- The section of concluding and considerations for implementation in lines 459 to 491 mainly repeated the results of data analysis and lacked considerations for implementation, especially the considerations of the value of these results for policy implementation, which needs to be rewritten in the revised version.
- We made a correction

Reviewer 2 Report
Thank you for this presentation which I read with interest.
The authors talked about elderly health and protective measures during the covid-19 pandemic in the opinion of the Polish society.
Line 1, The title of their manuscript is concise, specific and relevant.
ABSTRACT
Line 13, The abstract is the heading of the whole section. The abstract should be a single paragraph and should follow the style of structured abstracts, but without titles, so headings are eliminated.
Would you mind introducing the design and the main methods or treatments applied?
The abstract should be a total of about 200 words maximum.
Line 21-22 and 24-25 repeat the idea of negative attitudes.
Line 34 omit the adjective "vast".
INTRODUCTION
The introduction should briefly place the study in a broad context and highlight why it is crucial. It should carefully review the current state of the research field and cite key publications. Highlight controversial and divergent hypotheses where necessary. Keep the introduction understandable for scientists working outside the topic of the paper.
Line 51 omit individual countries from the end of the paragraph. It is redundant.
Line 53, please start the paragraph with "As the last pandemic has shown, ..." to omit line 57.
Line 128, please include the main objective of the work in the introduction, not in the methodology.
Line 134 put generically in the objective: "Due to its socio-demographic characteristics, knowledge of ageing and contact with older people in private and professional life, ...".
MATERIALS AND METHODS
Line 128 present the critical elements of the study design at the beginning of the paper, as This is a cross-sectional study that was carried out in the first week of February 2021.
Sample and participants
Line 141-143 merge them so as not to be redundant.
Indicate the eligibility criteria and the sources and methods of participant selection. These should be described in sufficient detail so that others can replicate and build on the published results: How were participants contacted, why is the sample known to be representative, was it purposive or randomised (following criteria of the proportion of age, sex, marital status, or other variables of the general population)?
Explain how the study size was arrived at.
Measures
What were the response options for the socio-demographic variables marital status, education, professional status, seniority, gross annual income, or were they open-ended questions?
Line 157, the FAQ instrument, was it validated, what were its psychometric properties?
Line 163, there is a typographical error with the inverted commas. Also, regarding health and protective measures for seniors during the covid 19, what was the procedure for creating the scale, was a pilot test carried out beforehand?
Line 171, put the name of the scale in italics, a proper name (the first letter in the lower case), and all scale names.
Line 175, please consider including section 2.3 Procedure and data collection at this point.
RESULTS
Although the description of the results with figures is fascinating, I consider that the results section is too long, and perhaps some results could be omitted. Provide a concise and accurate description of the experimental results. Assess the appropriateness of presenting the results in text, table and figures, without detracting from the discussion and conclusion. For example, I propose to delete figure 1.
TABLES
Format tables 1 - 4 in Vancouver style. All table columns should have an explanatory title. To facilitate the editing of larger tables, smaller fonts can be used, but no smaller than 8 pt. Authors should use the table option in Microsoft Word to create tables. Acronyms, abbreviations or initials should be defined the first time they appear in each of the three sections: the abstract, the main text, the first figure or table. The acronym/ abbreviation/initialism should be added in parentheses after the written form when defined for the first time, in the case of tables as a note.
Table 3. Format according to Vancouver standards. Avoid collapsing words if it is not necessary. The name of the columns is usually "N" for what you have called Freq. and "%" for what you have called "prop". If used, footnote the meaning as a Note. Change the comma to a complete stop for the description of decimals.
DISCUSSION
The authors should discuss the results and how they can be interpreted from previous studies and working hypotheses. The results and their implications should be discussed in the broadest possible context.
Would you please discuss further the limitations of the results?
Author Response
Response to Reviewer 2 Comments
Dear Reviewer,
We very much appreciate the careful review of our manuscript titled “Health and protective measures for seniors during the covid-19 pandemic in the opinion of polish society”. We have studied your comments carefully and made major corrections which we hope meet with your approval. We answer your questions or comments in detail in the following texts:
- ABSTRACT
- Line 13, The abstract is the heading of the whole section. The abstract should be a single paragraph and should follow the style of structured abstracts, but without titles, so headings are eliminated. Would you mind introducing the design and the main methods or treatments applied? The abstract should be a total of about 200 words maximum. Line 21-22 and 24-25 repeat the idea of negative attitudes. Line 34 omit the adjective "vast".
- As suggested, we made corrections.
- INTRODUCTION
- The introduction should briefly place the study in a broad context and highlight why it is crucial. It should carefully review the current state of the research field and cite key publications. Highlight controversial and divergent hypotheses where necessary. Keep the introduction understandable for scientists working outside the topic of the paper.
Line 51 omit individual countries from the end of the paragraph. It is redundant.
Line 53, please start the paragraph with "As the last pandemic has shown, ..." to omit line 57.
Line 128, please include the main objective of the work in the introduction, not in the methodology. Line 134 put generically in the objective: "Due to its socio-demographic characteristics, knowledge of ageing and contact with older people in private and professional life, ...".
- As suggested, we made corrections.
- MATERIALS AND METHODS:
- Sample and participants
- Line 141-143 merge them so as not to be redundant.
- Indicate the eligibility criteria and the sources and methods of participant selection. - As suggested, we made corrections.
- These should be described in sufficient detail so that others can replicate and build on the published results: How were participants contacted, why is the sample known to be representative, was it purposive or randomised (following criteria of the proportion of age, sex, marital status, or other variables of the general population)? - As suggested, we made corrections.
- Explain how the study size was arrived at. - The size of the survey was obtained by carrying it out online and using a special online platform - Survgo.
- Measures
- What were the response options for the socio-demographic variables marital status, education, professional status, seniority, gross annual income, or were they open-ended questions? - Please find all information in Table 3.
- Line 157, the FAQ instrument, was it validated, what were its psychometric properties - As suggested we added 2 articles [21,22]
- Line 163, there is a typographical error with the inverted commas. Also, regarding health and protective measures for seniors during the covid 19, what was the procedure for creating the scale, was a pilot test carried out beforehand? - As suggested, we made corrections.
- Line 171, put the name of the scale in italics, a proper name (the first letter in the lower case), and all scale names. As suggested, we made corrections.
- Line 175, please consider including section 2.3 Procedure and data collection at this point. As suggested, we added the “Procedure and data collection” section.
- RESULTS
- Although the description of the results with figures is fascinating, I consider that the results section is too long, and perhaps some results could be omitted. Provide a concise and accurate description of the experimental results. Assess the appropriateness of presenting the results in text, table and figures, without detracting from the discussion and conclusion. For example, I propose to delete figure 1. Thank you for your suggestion.
We discussed this topic in the team. Finally, we decided not to shorten this part.
- TABLES
- Format tables 1 - 4 in Vancouver style. All table columns should have an explanatory title. To facilitate the editing of larger tables, smaller fonts can be used, but no smaller than 8 pt. Authors should use the table option in Microsoft Word to create tables. Acronyms, abbreviations or initials should be defined the first time they appear in each of the three sections: the abstract, the main text, the first figure or table. The acronym/ abbreviation/initialism should be added in parentheses after the written form when defined for the first time, in the case of tables as a note.
- Table 3. Format according to Vancouver standards. Avoid collapsing words if it is not necessary. The name of the columns is usually "N" for what you have called Freq. and "%" for what you have called "prop". If used, footnote the meaning as a Note. As suggested, we made corrections.
Change the comma to a complete stop for the description of decimals. We discussed this topic in the team. Finally, we decided not to shorten this part.
- DISCUSSION
- The authors should discuss the results and how they can be interpreted from previous studies and working hypotheses. The results and their implications should be discussed in the broadest possible context. in progress/zrobione
Would you please discuss further the limitations of the results?
- As suggested, we made corrections.

Reviewer 3 Report
This is an interesting and important study under era of COVID-19 pandemic, but this article has many weakness.
- Authors recruited the participants through internet-based survey. Authors need to explain the survey method and environement of survey in detail.
- what was the predetermined the age, gender proportion for the survey? or was it just open survey without restriction to age, gender proportion?
- who made questionnaire. How did you decide on the questions?
- Is logistic regression a multivariate analysis? did you control with the confounding variables?
- Tables must be readable. for ex, on table 1. what gender was coded '1' .Authors describe in detail on the footnote about ex, Place of residence < 50 K, S/D/W, Vocational edu, Annual income
- on Table 4. what is Place of resi-dence Marital status [1]?
- what do you mean by 'activation programs for seniors'
- what do you mean by learners? students?
9. figures are too complicated . i would recomment the categorization of the answer options. for example, rather to definite yes --> yes.
Author Response
Response to Reviewer 3 Comments
Dear Reviewer,
We very much appreciate the careful review of our manuscript titled “Health and protective measures for seniors during the covid-19 pandemic in the opinion of polish society”. We have studied your comments carefully and made major corrections which we hope meet with your approval. We answer your questions or comments in detail in the following texts:
- Authors recruited the participants through internet-based survey. Authors need to explain the survey method and environement of survey in detail.
- As suggested, we made corrections.
- what was the predetermined the age, gender proportion for the survey? or was it just open survey without restriction to age, gender proportion?
- As suggested, we made corrections.
- who made questionnaire? How did you decide on the questions?
- A questionnaire was developed by the research team. It was prepared on the basis of previous discussions with health care specialists (including geriatricians, epidemiologists, physiotherapists and psychologists working with the elderly), whom we asked to identify the most important issues related to health and protective measures for seniors during the covid-19 pandemic. Then, the team conducted a qualitative pilot study, which allowed to verify the questions in terms of their understanding, readability and structure, on a group of 20 people of different sex, age and level of education.
- Is logistic regression a multivariate analysis? did you control with the confounding variables?
- Yes, all of the ordinal regressions presented in the text are multivariate. Each table depicts estimates from a single model; thus each effect was estimated with control of all the remaining predictors.
- Tables must be readable. for ex, on table 1. what gender was coded '1'. Authors describe in detail on the footnote about ex, Place of residence < 50 K, S/D/W, Vocational edu, Annual income
- on Table 4. what is Place of resi-dence Marital status [1]?
- Regarding questions 5 and 6: this was intentionally left out. The contrast coding is always somewhat arbitrary (e.g., whether females are coded as -1 or 1) and interpretation of these coefficients is much more complex for factors with more than 2 levels (like Marital status). Since we used sum-to-zero contrasts (which assures that together with scaled continuous predictors the coefficients represent effects around the 'center' of the data and also makes estimation procedure much more efficient) the coefficients for Marital status and Place of residence actually represent differences from the global mean for n-1 arbitrarily selected levels. On top of that, the coefficients are on logit scale, which makes the direct interpretation quite daunting. For these reasons we prefer to leave the tables as they are. Readers interested in the estimated proportions of answers can find them easily accessible in the figures corresponding to the tables. From these figures it is relatively easy to see what percentage of people (in each group of interest) provided which answer to a question, and based on these proportions to infer in which group the agreement level was higher (it's also directly indicated in the description of the results).
- what do you mean by 'activation programs for seniors'
- We changed into “Therapeutic activation programs” for elderly people. Those programs are dedicated for isolated seniors and focus on health promotion and wellness (e.g. tele rehab programs).
- what do you mean by learners? students?
- We changed into “students”.
- figures are too complicated. I would recomment the categorization of the answer options. for example, rather to definite yes --> yes.
- All of the models were estimated using ordinal regression (an extension of logistic regression) in which the dependent variable is treated as an ordered factor. The figures present predicted (from the model) proportions of each response, as a function of all of the predictors in the model. Hence, it would go against the chosen analytical method to merge the responses as proposed (especially that the model itself distinguishes between rather and definitely).

Round 2
Reviewer 3 Report
Well answered mostly.
on Table 4. what is Place of residence, gender? for ex, urban, men were coded as 1? Alternatively, authors just replace place of residence into urban residence, gender into men.
Author Response
Dear Reviewer, thank you for your opinion. Our answer is in the attachment.
Dear Reviewer,
Thank you for opinion. We answer your questions or comments in detail in the following texts:
- … on Table 4. what is Place of residence, gender? for ex, urban, men were coded as 1? Alternatively, authors just replace place of residence into urban residence, gender into men.
- Place of residence was treated as an ordered factor with linear contrast; thus, the coefficient represents the expected difference (on a logit scale) between any two consecutive levels of the factor.
As requested, the specific levels of the factors are now provided in the table instead of the numbers.
